# Endocannabinoid System and Metabolism: The Influences of Sex

**DOI:** 10.3390/ijms252211909

**Published:** 2024-11-06

**Authors:** Isabel Forner-Piquer, Christian Giommi, Fiorenza Sella, Marta Lombó, Nina Montik, Luisa Dalla Valle, Oliana Carnevali

**Affiliations:** 1Centre for Pollution Research and Policy, Department of Life Sciences, College of Health, Medicine and Life Sciences, Brunel University London, Uxbridge UB8 3PH, UK; iforner@icm.csic.es; 2Department of Life and Environmental Sciences, Polytechnic University of Marche, 60131 Ancona, Italy; c.giommi@staff.univpm.it (C.G.); f.sella@pm.univpm.it (F.S.); 3INBB—Biostructures and Biosystems National Institute, 00136 Roma, Italy; 4Department of Molecular Biology, Universidad de León, 24071 León, Spain; 5Department of Odontostomatological and Specialized Clinical Sciences, Polytechnic University of Marche, 60020 Ancona, Italy; nina.montik@ospedaliriuniti.marche.it; 6Department of Biology, University of Padua, 35131 Padova, Italy; luisa.dallavalle@unipd.it

**Keywords:** metabolic disorders, liver, appetite, sexual dimorphism, sex hormones, endocannabinoid system, CB1, lipid signaling

## Abstract

The endocannabinoid system (ECS) is a lipid signaling system involved in numerous physiological processes, such as endocrine homeostasis, appetite control, energy balance, and metabolism. The ECS comprises endocannabinoids, their cognate receptors, and the enzymatic machinery that tightly regulates their levels within tissues. This system has been identified in various organs, including the brain and liver, in multiple mammalian and non-mammalian species. However, information regarding the sex-specific regulation of the ECS remains limited, even though increasing evidence suggests that interactions between sex steroid hormones and the ECS may ultimately modulate hepatic metabolism and energy homeostasis. Within this framework, we will review the sexual dimorphism of the ECS in various animal models, providing evidence of the crosstalk between endocannabinoids and sex hormones via different metabolic pathways. Additionally, we will underscore the importance of understanding how endocrine-disrupting chemicals and exogenous cannabinoids influence ECS-dependent metabolic pathways in a sex-specific manner.

## 1. Introducing the ECS

The characterization of the psychotropic component of marijuana, Δ9-tetrahydrocannabinol (THC), in the 1960s led to the identification of high-affinity THC-binding sites in the brain, specifically the cannabinoid receptor type 1 (CB1), discovered in 1990. Three years later, in 1993, the cannabinoid receptor type 2 (CB2) was cloned. Both CB1 and CB2 are G-protein-coupled receptors (GPCRs) primarily found in neuronal and immune cells, respectively.

The discovery of cannabinoid receptors prompted an intensive search for their endogenous ligands. However, the isolation of the first endogenous mediator, anandamide (AEA), was not achieved until 1992 due to its lipophilic and highly unsaturated nature. This significant finding was followed by the identification of a second agonist, 2-arachidonoylglycerol (2-AG), in 1995. It was also in 1995 that the term “endocannabinoids” was proposed for these endogenously produced cannabinoids derived from arachidonic acid [1].

AEA and 2-AG belong to two different lipids family: these are the fatty acid amides known as *N*-acylethanolamines and the 2-acylglycerols, respectively. Endocannabinoids are produced and released “on demand”. This means that they are synthesized in a cell- and time-specific manner in response to physiological stimuli and pathological conditions in order to restore homeostasis [2,3]. Interestingly, recent reports have shown that endocannabinoids can also be stored in organelles [4]. The biological activity of endocannabinoids is further controlled by transporters that allow them to cross the cell membrane. However, the identification of the endocannabinoid’s membrane transporters remains elusive, but their activity has been proved, mostly for AEA [5]. A small number of studies are available regarding the uptake of 2-AG due to its rapid degradation. Molecular identification is still under debate and is expected to have major implications for therapeutic interventions [6]. In addition to these membrane transporters, endocannabinoids can be transported via passive diffusion or the exocytosis of microvesicles, although passive diffusion is less likely due to the polar nature of endocannabinoids. Due to their lipid nature, endocannabinoids also require appropriate carriers, AEA- and 2-AG-binding proteins, to move within the cytosol. These intracellular transporters were extensively reviewed in [2]. To name some, albumin and fatty acid binding proteins (FABPs) are involved in intracellular traffic for both AEA and 2-AG. For AEA, this involves heat shock protein 70 (HSP70), FAAH-like AEA transporter (FLAT), and sterol carrier protein 2 (SCP2), and retinol-binding protein 2 (RBP2) for 2-AG. It is worth noting that FABP1, albumin, and FLAT are mainly localized in the liver [2,4].

Endocannabinoids function as retrograde signaling molecules. Neuronal stimulation induces the synthesis of cannabinoids in the postsynaptic cells, allowing them to travel backward across the synapse to activate CB1 receptors on presynaptic neurons [2,3]. CB1 activation initiates various responses, including the inhibition of adenylyl cyclase activity, the activation of mitogen-activated protein kinases (MAPKs), and the modulation of ion channels to restore homeostasis and control neuronal excitability [2,3]. Nonetheless, non-retrograde endocannabinoid signaling has also been reported (reviewed in [7]).

A few years later, the characterization of enzymes responsible for endocannabinoid biosynthesis and metabolism marked another breakthrough. Briefly, as it has been extensively reviewed [3,8,9], the ECS enzymatic machinery consists of three biosynthetic enzymes that convert the respective precursors into AEA and 2-AG: these are *N*-acyl-phosphatidylethanolamine-hydrolyzing phospholipase D (NAPE-PLD) and sn-1-specific diacylglycerol lipases (DAGLα and DAGLβ), respectively. For degradation, fatty acid amide hydrolase (FAAH) is the primary enzyme responsible for AEA and, to some extent, 2-AG hydrolysis. 2-AG is also metabolized by monoacylglycerol lipase (MGLL). These enzymatic pathways are crucial for regulating endocannabinoid levels.

The aforementioned ECS components (the signaling lipids AEA and 2-AG, their metabolic enzymes, and CB1 and CB2 receptors) comprise the “classic ECS”. Recently, it has been discovered that the ECS is more complex than initially understood. The ECS encompasses a broader endocannabinoid-related network known as the “expanded ECS” or endocannabinoidome, which includes additional bioactive lipid derivatives, receptors, and alternative enzymatic pathways [8,10,11,12]. The existence of the endocannabinoidome raises new questions and complicates the biological implications of the ECS, which also interplays with other signaling mechanisms and regulatory pathways [3]:

(1) There is interaction with a broad and overlapping range of molecular targets beyond the classical receptors, such as nuclear receptors or ion channels, resulting in non-CB1- or non-CB2-mediated effects. (2) The endocannabinoid receptors can exhibit distinct roles during pathological conditions. (3) Redundant biosynthetic and catabolic pathways for AEA and 2-AG exist, utilizing alternative routes and other lipid mediators, therefore, sharing enzymatic pathways with other mediators. (4) There is an overlap and interaction with other pathways, as exemplified by the role of 2-AG as a crucial intermediate in lipid metabolism. (5) The allosteric modulation of CB1 and CB2 by other endogenous ligands, such as peptides or sex steroids and their precursors, can interact to enhance or inhibit CB1 activation. Consequently, since endocannabinoids are not the only receptor agonists, these other ligands may induce distinct receptor conformations and ultimately stimulate different pathways. (6) Endocannabinoidome mediators may act in synergy or competition with endocannabinoids [2,3,8,10,13,14].

## 2. Sex Differences in the ECS

The interaction between sex hormones and the ECS is complex and involves bidirectional regulation. Endocannabinoids can modulate the levels of sex steroids and gonadotropins; in turn, the ECS is strongly influenced by sex hormones (reviewed in [15]). Along these lines, there is an ongoing debate about the ECS–steroid interface and the possibility that hormone-dependent ECS actions might also be tissue- and time-specific [16,17]. This debate is supported by the significant overlap in the molecular pathways of endocannabinoids and estrogen (reviewed in [18]). Furthermore, sexual dimorphism in the ECS is already evident in early development (reviewed by [19]).

Gorzalka and Dang [20] suggested that, unlike the classical negative feedback in the hypothalamic–pituitary–gonadal (HPG) axis, the ECS appears to modulate the release of androgens and estrogen by negatively affecting gonadotropin-releasing hormone (GnRH) secretion from the hypothalamus, consequently downregulating the production of luteinizing hormones (LHs) from the anterior pituitary. Accordingly, AEA, 2-AG, FAAH, CB1, and CB2 have been detected in hypothalamic GnRH neurons [21]. Current theories propose that the ECS modulates GnRH by inhibiting the neuronal systems involved in GnRH circuitry, release, and transcription, while also activating inhibitors of GnRH activity (reviewed by [22,23]). Specifically, the postsynaptic release of endocannabinoids acts at the presynaptic level to inhibit the release of specific neurotransmitters (i.e., gamma-aminobutyric acid (GABA), glutamate) involved in GnRH secretion. GABAergic transmission plays a key role in GnRH’s secretory activity. In male mice, retrograde endocannabinoid signaling reduces GABAergic transmission to GnRH neurons. Specifically, GnRH neurons release 2-AG, which activates CB1 on the afferent GABAergic neurons. As a result, postsynaptic GABA receptors present on GnRH-secreting neurons are not activated, thereby reducing GnRH release [24].

Based on the understanding that these circuits can be modified by sex and sex steroid feedback [22], and that exogenous estradiol (E2) administration inhibits GnRH neuron firing [25], notable findings have emerged. E2 mediates the reduction in GnRH neuronal activity through 2-AG/CB1 retrograde signaling in female mice, with the E2-induced effects attenuated by a CB1 inverse agonist and an endocannabinoid synthesis blocker. Estrogen receptor beta (ERβ) is required to achieve this rapid effect in GnRH neurons [25]. Similarly, Glanowska and Moenter reported that the local feedback modulation of GnRH is regulated by the ECS in a steroid-dependent manner [26]. Hypothalamic *Gnrh* gene expression is also lower in CB1 knockout (CB1 KO) male mice, although this phenotype is not restored via external E2 administration. In contrast, different results were observed for GnRH receptor (Gnrhr) levels in the pituitary [27]. Beyond GnRH, endocannabinoids can regulate other signaling systems by acting on the glutamatergic neurons, adding further complexity [28,29]. Huang and Woolley showed that acute treatment with estradiol (E2) suppresses inhibitory GABAergic neurotransmission in the hippocampus of ovariectomized (OVX) rats. In their study, E2 acted through estrogen receptor alpha (ERα) to stimulate postsynaptic metabotropic glutamate receptor (mGluR1)-dependent AEA mobilization, which, in turn, retrogradely decreased GABA release in female rats, but not in males. Interestingly, the same group also reported that E2 potentiates excitatory synaptic transmission by increasing glutamate release in the hippocampus of female rats via estrogen receptor beta (ERβ) [30]. The authors suggested that these in vivo effects may likely be triggered by locally synthesized E2 [31]. Additionally, the same group showed that, independently of E2, a FAAH inhibitor suppressed GABAergic inhibitory synapses in the hippocampus of gonadectomized females but not in males, indicating AEA production and suggesting that a significant fraction of those inhibitory synapses may be regulated by AEA or another FAAH-related mechanism in females. These studies demonstrated sexual dimorphism in both the E2-dependent and E2-independent regulation of the ECS [32]. Similarly, Scorticati and colleagues also showed the inhibitory effects of AEA on GABA release in OVX and estrogen-primed (OVX-E) rat hypothalamic explants [33]. Additionally, the ECS may modulate GnRH neurons via kisspeptins (reviewed by [22,23]).

In this regard, FAAH is another point of interaction between the ECS and estrogens. Response elements for estrogen were identified in the *Faah* sequence a few years ago [34]. A study by Maccarrone and colleagues highlighted that E2 decreased FAAH activity in the murine uterus [35]. Consistent with this, E2 modulated NAPE-PLD and FAAH activity in the rat brain [36]. The synthesis of AEA was significantly higher in the hypothalamus of OVX-E-primed rats compared to OVX rats or males [33]. Moreover, Maia and colleagues (2017) [37] reported a significant increase in NAPE-PLD and cyclooxygenase-2 (COX-2) at both the gene and protein levels following E2 administration in the uterus of OVX rats. It is noteworthy that the ECS activity and CB1 density in the central nervous system (CNS) are not static and fluctuate throughout the estrous cycle, with findings differing between humans and other species (reviewed in [20]). In rodents, the expression of specific elements of the ECS is modulated by estrogen and other sex hormones in reproductive tissues [4,37]. However, E2 did not modify CB1 protein levels in OVX and OVX-E rats, although males showed lower CB1 expression than females [33]. The plasma levels of LH were similarly suppressed in OVX rats treated with AEA and CB1 antagonists, with this effect reversed in OVX-E-primed rats [33].

At the central level, 2-AG and AEA differ between males and females and fluctuate during the ovarian cycle, similar to CB1 binding affinity and CB1 density in different brain areas (reviewed in [19,38,39]). As expected, the relative expression of endocannabinoid enzymes also diverges in specific brain regions of adolescent male and female rats [40]. Farquhar and colleagues reported that vehicle-treated intact adult rats exhibited minimal sex differences in terms of CB1 density. The authors acknowledged that these contrasting results across studies could be attributed to the diverse methodologies used, highlighting the need for standardized protocols [41]. Castrated males and OVX female rats showed region-specific reductions in CB1, providing a potential explanation for sex-related differences in the central ECS and their effects [18]. The administration of E2 to OVX rats reduced CB1 density in specific brain areas [42]. Supporting this, the gene coding for CB1, *Cnr1*, in the anterior pituitary area was diminished in castrated males, and this situation was not improved after testosterone treatment [38].

Serum testosterone levels were reduced along with low serum LH and E2 in CB1 KO male mice [27,43]. Additionally, AEA administration decreased serum testosterone and serum LH levels in both wild-type and CB1 KO male mice [43]. Generally, in animal models, endocannabinoids decrease testosterone levels by inhibiting its synthesis, accelerating its metabolism, and impairing the binding of androgen to the androgen receptor (AR) [44].

Due to its density and distribution, CB2 has been less extensively explored than CB1. Several in vitro studies have been published regarding the binding capacity of selective ER modulators to CB2 [45,46], suggesting that CB2 could be a potential therapeutic target for treating different pathologies [18,47].

Progesterone also regulates ECS signaling by controlling FAAH [35]. In the mouse uterus, progesterone and/or estrogen downregulated the gene expression of *Nape-pld* [48]. Pregnenolone, a precursor of sex steroids, acts as a negative allosteric modulator of CB1, inhibiting CB1-specific signaling, while its metabolites serve as positive allosteric modulators of the GABA_A_ receptor (reviewed in [28]). However, mixed effects have been observed regarding CB1 density in different brain areas of female rats [49]. Sabatucci and colleagues showed that pregnenolone was the most effective steroid in terms of enhancing FAAH membrane affinity in rats. The authors suggested that pregnenolone may induce a change in protein conformation, improving membrane binding affinity [17]. In addition to these findings in mammals, the AEA-dependent modulation of GnRH-secreting neurons was described in the amphibian *Rana esculenta* at the central level via CB1 [50]. In sexually immature goldfish (*Carassius auratus*), an intraperitoneal injection of 2.5 mg/kg body weight of E2 did not affect brain *cnr1* or *cnr2* expression; however, *faah* mRNA levels were upregulated [51]. The treatment of juvenile sole (*Solea solea*) with 10^−8^ M of E2 induced *cnr1* gene expression in the brain [52]. Nevertheless, the available experimental data are insufficient to draw definitive conclusions about hormonal–ECS interactions in non-mammalian models.

In summary, hormone-driven fluctuations in ECS receptors, endocannabinoids, and enzymatic activity provide a basis for understanding sex differences in endocannabinoid-related functions, such as metabolic functions and energy homeostasis. Based on these premises, we will next review the experimental evidence associated with the sexual dimorphism involved in ECS-dependent hepatic metabolism, aiming to provide a general overview of the crosstalk between ECS and sex steroids. Within this framework, we conducted a PubMed and Google Scholar search, using combinations of different keywords related to the ECS, sex steroids, hepatic metabolism, food intake, metabolic dysfunctions, and endocannabinoids. Our review was limited to data obtained in vivo from different experimental models (i.e., murine and teleost models). We did not specify a timeframe for the bibliographic search, but the latest peer-reviewed studies were included in this review.

## 3. Involvement of the ECS in the Appetite Control

### 3.1. ECS Interactions with the Orexigenic and Anorexigenic Signals

The regulation of appetite is an intricate bidirectional process that requires cooperation between the CNS and peripheral organs, such as the pancreas, gut, liver, muscles, and adipose tissue. All these organs communicate through various systemic and paracrine signals. At the peripheral level, adipose tissue and the pancreas produce two of the main anorexigenic signals: these are leptin and insulin, respectively. These hormones work in conjunction with cholecystokinin, nesfatin, glucagon-like peptide-1 (GLP-1), and oxyntomodulin, which are produced in the gut, to reduce food intake and appetite [53,54]. Once released into the bloodstream, these anorexigenic signals reach the CNS and act on the catabolic neurons of the arcuate nucleus (ARC), promoting the production of α-melanocyte stimulating hormone (α-MSH) while inhibiting the anabolic neurons of the ARC [1,53]. Conversely, the intestine produces ghrelin, the main orexigenic hormone, which stimulates food intake and weight gain [53,55]. Ghrelin activates neuropeptide Y (NPY)- and agouti-related peptide neurons, leading to the release of GABA and norepinephrine, which subsequently increases appetite [53].

In recent years, the increasing incidence of obesity and metabolic disorders has prompted research into new treatments to regulate food intake and appetite [54]. Meanwhile, the role of the ECS in appetite regulation has emerged as a significant area of study [56,57,58]. CB1 is abundantly expressed in key brain regions involved in appetite regulation, such as the hypothalamus and limbic system. At the central level, the ECS stimulates appetite and enhances the palatability of food through the activation of CB1 by endocannabinoids [59,60,61]. Moreover, the levels of 2-AG positively correlate with ghrelin levels, and together with AEA, they increase hunger and food intake through CB1 binding [59]. AEA is one of the major players in appetite regulation, acting as an orexigenic compound. This primarily occurs through its interaction with CB1 [62,63,64]. This correlation was observed in mice through the administration of CB1 reverse agonists, SR141716A and AM251, which effectively neutralized the endocannabinoid-induced effects on food consumption. This result was also supported using CB1 KO mice [65,66]. The activation of CB1 in the paraventricular nucleus inhibits the release of 5-hydroxytryptamine (5-HT) and the subsequent serotonergic activity. This mechanism, along with the inhibition of GABA release, leads to food intake stimulation [67]. In contrast, the absence of CB1 has been linked to hypophagia and leanness in mice, resulting in reduced spontaneous caloric intake and body weight without affecting locomotor activity, body temperature, or energy expenditure [68].

As for CB2, its role in the regulation of food intake in rodents was recently reviewed by Rodríguez-Serrano and Chávez-Hernández [69]. CB2 is present in the mesocorticolimbic system, a brain area involved in the reward pathway. The activation of CB2 in dopamine neurons inhibits dopamine-induced neuronal firing in mice and modulates the hedonic reward associated with food consumption. However, CB2 ablation stimulates food intake and body weight gain, and also induces the hypertrophy of adipose tissue in mice. On the other hand, CB2 overexpression in the brain reduces food intake and increases glycemia. This difference has sparked growing interest in CB2 agonists as potential therapeutic targets for obesity treatment.

In addition, endocannabinoid-like compounds, such as the *N*-acyl derivative of palmitic acid and the *N*- and O-acyl derivatives of oleic acid, share metabolic enzymes with endocannabinoids and potentiate their effects. However, these compounds do not act through ECS receptors but exhibit agonistic activities for other receptors, such as the peroxisome proliferator-activated receptor α (PPARα), which modulates appetite and food intake [70,71]. One example is oleoylethanolamine (OEA), which is produced in the duodenum. OEA activates the nucleus tractus solitarius, decreases body weight, increases inter-meal latency, and affects energy expenditure by enhancing lipid utilization [72]. Accordingly, the anorectic effect of OEA administration in male rats was not suppressed following SR141716A and SR144528 treatments (CB1 and CB2 antagonists), supporting the idea that OEA does not act via the classical endocannabinoid receptors [59]. The administration of another endocannabinoid-like compound—palmitoylethanolamide (POEA), which is structurally related to OEA—to OVX rats demonstrated its anorexigenic capacity by modulating hypothalamic leptin signaling and reducing food intake, fat mass, and body weight [73]. Recently, the anti-obesogenic effects of OEA and POEA were assessed using male rats fed on a high-fat diet (HFD), revealing that both compounds were able to reduce HFD-induced weight gain, liver steatosis, inflammation, and dyslipidemia [74].

### 3.2. Sex Difference in the ECS-Mediated Appetite Control

As mentioned above, the ECS is influenced by sex steroids, and vice versa. Sexual dimorphism can be observed in terms of differently expressed ECS components and in the tissue levels of endocannabinoids. Considering the importance of the ECS in energy balance and appetite regulation, as well as its potential sexual dimorphism [75,76,77], we will now examine recent evidence regarding the sex-based role of the ECS in food intake and energy homeostasis.

Initial studies on food intake and food rewards were exclusively conducted using male mice [78,79,80]. These studies described that the genetic inactivation of CB1 diminished the operant response to palatable food, a finding further validated by the work of Ward and Dykstra [81]. Over time, male CB1 KO animals maintained an operant response to a high-sugar diet (HSD) for longer. Interestingly, while male CB1 KO mice were able to perform self-administration of corn oil (a fat reinforcer used to create an HFD) similarly to wild-type animals, this response was only maintained after the experiment in the CB1 KO male mice. These results indicate that although CB1 ablation does not affect the acquisition of HFD self-administration, it may be involved in maintaining this behavior over time. To further investigate this point, wild-type mice were treated with the CB1 antagonist SR141716A and the CB1 agonist CP-55940. The SR141716A treatment reduced the time required to develop an operant response to HSDs in both CB1 KO and pretreated wild-type mice compared to wild-type control animals. Moreover, SR141716A decreased the time to acquire an operant response to HFDs in the pretreated wild-type group. In the case of the CP-55940, this CB1 agonist significantly increased the operant response to HFDs in pretreated wild-type animals and produced a small but not significant increase in response to HSDs [81]. This highlights the greater involvement of CB1 in the operant response to HSDs compared to HFDs. Additionally, treatment with a CB1 antagonist and agonist effectively decreased or increased, respectively, the operant responses observed in these studies.

A subsequent study by the same authors [82] evaluated the reinstatement of operant responses to HSDs and HFDs, which refers to the restoration of a previously extinguished behavior upon the reintroduction of the original reinforcing stimulus after a period of extinction. The authors demonstrated that 1.0 and 3.0 mg/kg of a CB1 antagonist (SR141716A) were able to decrease HSD-seeking reinstatement. However, the antagonist did not affect the response to HFDs. Instead, this required a ten-fold higher concentration of SR141716A. This experimental setup further confirmed the role of CB1 in the response to sugar-rich foods compared to fat-rich foods in males. In agreement with the work of Ward and Dykstra [81], the CB1 KO group exhibited a slight decrease in HSD-seeking behavior without affecting the reinstatement of HFDs. Departing from these differences, the same research group assessed the capacity of both females and males (wild-type and CB1 KO mice) to perform the self-administration of palatable food using a corn oil HFD [83]. Interestingly, wild-type females and males developed this capacity simultaneously, however the ability of female CB1 KO to perform the self-administration of palatable food was delayed compared to both male CB1 KO and wild-type females. This suggests that the inactivation of CB1 in females may negatively impact their capacity for food intake, making them less prone to develop an addiction to fat-rich food. Surprisingly, no significant differences were observed between wild-type and CB1 KO males.

Regarding the maintenance of the self-administration of HFD in females and males, males were able to maintain it for a prolonged period compared to females although no differences were observed on the CB1 genotype. In the same study [83], the acquisition and maintenance of HSD self-administration in female and male mice was also investigated. The results confirmed previous observations where CB1 KO males exhibited delayed acquisition of HSD self-administration and maintained this capacity for a shorter time compared to the wild-type littermates. In contrast, in females, CB1 genetic deletion did not affect the acquisition time of HSD self-administration, but the maintenance of this behavior was shorter in wild-type females than in wild-type males. This evidence, together with the results from wild-type and CB1 KO females, suggests that females are less predisposed to develop an addiction to sugar-rich food than males. Therefore, the evidence reported by Ward and Walker [83] indicates that while males are more predisposed to acquire the self-administration for HSDs, females are more prone to develop the self-administration for HFDs. However, no information was reported at the cellular or molecular level to explain this sex-based response of the ECS.

Follow-up studies investigating the mechanisms underlying sexual dimorphism in the ECS at the brain level provided interesting insights using guinea pigs (*Cavia porcellus*). Male guinea pigs were found to be more sensitive to the hyperphagic effects of ECS activation than females [84,85]. When treated with a CB1 agonist (WIN 55,212-2), food intake, meal size and meal duration were improved in males. Conversely, the administration of the CB1 antagonist (AM251) reduced these parameters, with more pronounced effects in males. To investigate the cellular mechanisms, *Cavia porcellus* ex vivo brain slices were used to monitor miniature excitatory postsynaptic potentials (EPSPs) and inhibitory postsynaptic currents (IPSPs) in arcuate neurons. This experimental setup revealed a higher baseline EPSP value in female proopiomelanocortin (POMC) neurons compared to males.

POMC neurons located in the ARC of the hypothalamus are critical for regulating appetite and energy expenditure by producing peptides such as α-MSH, which suppresses appetite, and β-endorphin, which stimulates appetite [86,87]. When POMC neurons are activated, they promote satiety and reduce food intake. However, CB1 agonists selectively increase β-endorphin secretion in the neurons of the paraventricular nucleus, a hypothalamic region implicated in feeding behavior, without affecting α-MSH secretion [86,87]. Based on the findings of Diaz and colleagues [84], Kellert and co-workers [88] investigated how estrogen impacts cannabinoid-mediated changes in appetite and POMC neuronal activity in guinea pigs. E2 administration reduced the cannabinoid-induced presynaptic inhibition of glutamatergic inputs to POMC neurons [88], thereby limiting the influence of CB1 signaling on food intake. This effect was primarily mediated by the activation of ERα and the Gq-coupled membrane ER in females, as demonstrated by Washburn and co-workers [89]. Supporting this, E2 treatment also decreases CB1 levels in the hypothalamus of OVX guinea pigs [89,90,91]. In contrast, the presence of testosterone in males produced the opposite effect, enhancing the ECS-mediated inhibition of EPSPs in POMC neurons [91].

Testosterone present in males binds to the ARs on POMC neurons of the ARC, triggering the synthesis of 2-AG through DAGLα. This process increases intracellular calcium and activates calmodulin-dependent protein kinase and the AMP-activated protein kinase (AMPK) complex [92,93,94]. AMPK, which is sensitive to the AMP:ATP and ADP:ATP ratios, promotes the consumption of high-energy foods to restore energy balance. It is typically activated by orexigenic hormones like ghrelin [95] and inhibited by anorexigenic signals like leptin [96]. Once synthesized, 2-AG can retrogradely activate presynaptic CB1 receptors in the hypothalamic ventromedial nucleus (VMN) neurons that express steroidogenic factor 1 (SF-1), which is involved in appetite regulation and energy expenditure [97]. The VMN is a brain region that regulates various physiological processes, including feeding behavior, energy balance, and reproductive function [98].

The activation of CB1 depletes calcium levels and inhibits glutamate release, causing hyperphagia. Such stimulation in food intake hampers the signaling pathway involving phosphoinositide 3-kinases (PI3K), which produces phosphatidylinositol triphosphate (PIP3) and ultimately, activates the protein kinase B (Akt) in POMC neurons. This is a critical signaling cascade that regulates various functions, including cell growth, proliferation, survival, and metabolism [99]. Then, the endocannabinoid tone is once again elevated at the synapse, contributing to hyperphagia [92,93,94]. Results in male guinea pigs are consistent with those obtained in mice, where the activation of the ECS promotes the operant response to HSD [83]. Additionally, the antagonism or genetic deletion of CB1 was able to prevent this operant response in male mice but not in females [83]. In females, E2 negatively affects the synaptic regulation of VMN SF-1 neurons and ARC POMC neurons. The binding of E2 to ERα in POMC neurons triggers the release of nitric oxide (NO), synthetized by the nitric oxide synthases (nNOS). This, in turn, induces the catabolism of endocannabinoids, decreasing their synthesis and promoting their reuptake. Furthermore, NO may uncouple CB1 from endocannabinoids and prevent glutamate release from VMN SF-1 neurons [90]. The negative regulation of food intake by E2 in females is also dependent on the estrous cycle, with a weakened retrograde inhibition during the proestrus and diestrus phases compared to the estrus phase [94,100]. The information regarding endocannabinoid signaling that occur at the VMN SF-1/ARC POMC synapses in females and males were summarized in Figure 1.

The influence of HFD on the ECS and appetite regulation during gestation have evidenced notable sex-specific effects in the offspring. In a study where F0 female rats were fed with an HFD (28% fat) for eight weeks before mating and throughout gestation and lactation, male and female offspring exhibited distinct ECS-related changes. In male F1, maternal HFD exposure led to an increase in hypothalamic CB1, orexin-A (a neuropeptide involved in food intake regulation), and POMC, the precursor of α-MSH and β-endorphin, in POMC neurons. In contrast, CB1 and CB2 were unchanged and upregulated, respectively, in female F1 hypothalamus [101].

Concerning the effects of a HFD on the ECS at the peripheral level, sex-specific alterations were also observed in the F1. In male offspring, maternal HFD treatment resulted in reduced CB1 and increased FAAH levels in brown adipose tissue (BAT), while females exhibited improved levels of CB2 and MGLL. Although the reduction of the CB1 in males could suggest higher thermogenic activity in BAT, the analysis of thermogenic markers did not show any changes [101]. Since the response of CB1 at the hypothalamic level is correlated with a reduction in the sympathetic nervous system tone, the authors suggested a possible compensatory mechanism that maintains the thermogenesis of BAT. Additionally, the downregulation of the gene coding for the deiodinase 2 (*Dio2*), along with the increased levels of triiodothyronine (T3), were observed in female pups. T3 has the potential to stimulate the function of uncoupling protein 1 (UCP-1), also known as thermogenin, which is a mitochondrial carrier protein found in BAT. Interestingly, the levels of UCP-1 remained unaltered in female pups, resulting in the absence of thermogenesis dysregulation. This suggests that there is a possible compensatory mechanism in females as well. Despite the presence of sex-specific ECS variations at both central and peripheral levels regarding energy balance, rat offspring showed an increment in body weight and white adipose tissue (WAT), together with the preference for HFD in adulthood in both sexes. Regardless of the alterations in FAAH in males and MGLL in females, one limitation of the aforementioned study was the absence of the endocannabinoid levels.

To further investigate the molecular mechanisms involved in the hypothalamic ECS, the same research group conducted transcriptomic analyses to assess the effect of maternal HFD. The authors observed the differential expression of *Faah* and *Cnr1* in male and female pups. The authors suggested that these differences may result from potential interferences with other regulatory systems, such as microRNAs; changes in estrogen signaling and increased AR binding to the *Cnr1* promoter [102]. Using a similar experimental approach, the influence of maternal HFD (29% fat) administration on the levels of endocannabinoids in breast milk in rats was investigated [103]. The offspring of both sexes showed an increase in WAT on postnatal day 21. Maternal HFD reduced AEA and 2-AG levels in milk and induced sex-specific changes in the nucleus accumbens. In male weanling rats, CB1 and CB2 were increased and FAAH decreased, while in females a reduction in MGLL was observed. The authors also analyzed dopamine signaling, given that endocannabinoids can inhibit GABA inputs to dopamine neurons, increase dopamine release and promote palatable-food-seeking behavior. Notably, the molecular findings related to HFD preference were solely observed in male pups. Female offspring did not demonstrate any food preference, despite alterations in the ECS and dopamine signaling [101].

Maternal food restriction and its effects on the offspring’s ECS, food preference, and feeding behavior were recently assessed in rats [104]. In male offspring, maternal food restriction minimized the levels of arachidonic acid and OEA in the hypothalamus, but did not affect the tone of 2-AG and AEA. In contrast, lower levels of 2-AG and POEA were observed in female offspring in response to maternal food restriction. At the hippocampal level, male offspring showed increased levels of 2-AG and arachidonic acid, whereas in females, AEA, arachidonic acid, and POEA levels were significantly higher. Interestingly, only females exhibited lower levels of AEA, arachidonic acid, and POEA in the olfactory bulb. These findings suggest that there is a sex-specific alteration in the endocannabinoid and endocannabinoid-like profiles of the offspring of rats following moderate maternal diet restriction, highlighting the potential role of the ECS in the long-lasting effects of undernutrition and low body weight during early development. In this regard, early weaning in rats triggered sex-specific alterations in food preference. Male rats preferred HFDs during the first 30 min of the trial and HSDs after 12 h, while females showed no preference between the two diets. These behavioral phenotypes were associated with a reduction in FAAH and MGLL in the lateral hypothalamus of early-weaning males, consistent with the reported hyperphagia, as an elevated central endocannabinoid tone stimulates food intake. On the other hand, early-weaning females revealed decreased NAPE-PLD and raised FAAH, but no preference for palatable diets. The authors suggested that such unexpected results might be related to a decline in dopaminergic activity, as dopaminergic signaling inhibits feeding behavior. These sex-specific differences partially explain the exacerbated food consumption in early-weaning animals, with a particular preference for palatable food only observed in males [105].

## 4. Involvement of the ECS in Lipid Metabolism

### 4.1. ECS Interactions with Metabolism

Considering the physiological implications of the ECS, its role has been intensively investigated in several disorders, such as inflammatory diseases, obesity, metabolic syndrome, and hepatic dysfunctions. One of the most common metabolic disorders is non-Alcoholic Fatty Liver Disease (NAFLD), characterized by the accumulation of fat in the liver and the development of insulin resistance [106,107]. Exacerbated adipocyte lipolysis and high circulating levels of free fatty acids are the main contributors to fat accumulation and steatosis [108]. In the mouse model, the overactivation of CB1 is related to obesity-associated fatty liver and insulin resistance, two major conditions correlated with NAFLD. The activation of CB1 stimulates lipogenesis pathways, promotes the accumulation of mono-unsaturated fatty acids, and inhibits FAAH production in the liver, increasing the AEA tone and leading to insulin resistance (reviewed in [109]). CB1 is associated with the upregulation of genes coding for the lipogenic transcription factor SREBP-1c and its targets, acetyl-CoA carboxylase-1 and fatty acid synthase (FAS), resulting in hepatic fibrosis, steatosis, and lipid accumulation [110]. Experimental studies utilizing the peripheral CB1 antagonist AM6545 have shown metabolic improvements and the amelioration of hepatic steatosis in diet-induced obese male mice [111]. Similarly, in the teleost *Danio rerio* larvae, the administration of AEA upregulated the gene expression of *cnr1* and *srebp*, which promote fatty acid synthesis and cholesterol synthesis and uptake [112]. Accordingly, when *cnr1* was overexpressed in the liver, the transcripts of genes involved in fatty acid production, transport, and storage were increased, causing hepatic lipid accumulation in both zebrafish larvae and adults [113].

In addition to CB1, *Cnr2* expression was also enhanced in the livers of male HFD-fed wild-type mice [114]. Treatment with JWH-133, a CB2 agonist, exacerbated liver inflammation and the development of fatty liver in HFD-fed wild-type mice, suggesting a potential association between CB2 and an increased risk of developing NAFLD. In line with this, HFD-fed male *Cnr2^−/−^* mice exhibited minimal steatosis and reduced liver triglyceride (TG) concentrations [114]. However, the role of CB2 in liver diseases remains controversial. Opposite outcomes have been observed with other CB2 agonists, such as JWH-015 and HU308, both of which appear to improve CB2-mediated anti-obesity signaling and prevent metabolic disease from developing in this animal model [115,116]. The differing therapeutic outcomes may be attributed to variations in off-target drug effects.

### 4.2. Sex Differences in Lipid Metabolism

Considering the expression of ERα and ERβ in both the liver and adipose tissue, there is growing interest in deciphering the mechanisms by which estrogen regulates lipid homeostasis. As previously mentioned, estrogens are involved in regulating food intake, body weight, glucose homeostasis/insulin sensitivity, and lipolysis/lipogenesis [117]. Estrogen and testosterone play protective roles in preventing fatty liver in rodent models [118]. To study the contributions of E2 to HFD-induced NAFLD, orchidectomized (OXR) male Sprague Dawley rats were used as models. E2 treatment reduced liver lipogenesis, followed by the phosphorylation of acetyl coenzyme A carboxylase (ACC) via ERα, an enzyme involved in de novo fatty acid synthesis [119]. ERα KO male and female mice exhibited exacerbated adiposity, insulin resistance, and glucose intolerance, suggesting the critical role of this receptor in the WAT of both sexes [120]. Similarly, the brain-specific deletion of ERα has been associated with increased abdominal fat due to hypometabolism and energy balance dysregulation in female mice. The deletion of ERα in the CNS of male mice also promoted body weight gain [121]. ERα KO females were fatter than wild-type mice or ERβ KO, indicating that ERα plays a key role in protecting against metabolic dysfunction. OVX-ERα KO did not show an increase in adiposity, unlike in intact animals or ERβ KO, whereas OVX-ERβ KO mice exhibited adiposity and insulin resistance [122]. This suggests an anti-obesogenic role for ERβ, whose action has been reported to be tissue-specific and sex-dependent [123].

Regarding the contribution of the AR pathway to the regulation of hepatic homeostasis, 5α-dihydrotestosterone (DHT) activates the AR, leading to increased β-oxidation and a reduction in lipid accumulation and cholesterol biosynthesis in male rats. Additionally, the study described the protective effects of E2 and DHT on liver metabolism, as indicated by the reversed phenotype observed in hepatic histology [119]. Hepatic fat accumulation was also reported in OXR mice but not in OXR mice supplemented with testosterone [124]. Conversely, female mice treated with a low dose of DHT exhibited exacerbated de novo lipogenesis and increased hepatic lipid content [125]. In this context, the deficiency of hepatic AR resulted in liver steatosis in HFD-fed male mice, but not in females. AR-null male mice showed the upregulation of genes associated with de novo fatty acid synthesis markers (*Acc*, *Scd1* [stearoyl-CoA desaturase-1], *Srebp1c*, *Pparγ*) and a reduction in transcripts related to lipid oxidation (Malonyl-CoA decarboxylase, *Mcad*) [126]. Similar results were reported in studies using OXR and AR-depleted male mice, as well as in male HFD-fed animals, which exhibited higher levels of hepatic TGs relative to body fat. However, this phenotype was not observed when the animals were fed on regular chow [118,127]. Using different mouse strains, such as C57BL/6 and LKO, Norheim and colleagues found that male mice accumulate more hepatic TGs than females [118].

Furthermore, hepatic metabolism can be affected in a sex-dependent manner in mouse offspring after HFD feeding during the perinatal period. Maternal obesity leads to severe metabolic dysfunction, predominantly affecting F1 males compared to females. Additionally, the compositions of adipose tissue and hepatic TGs were modified differently depending on the sex of the offspring. Savva and colleagues speculate that females may possess specific regulators that protect against metabolic perturbations [128]. A follow-up study by the same group reported sex-specific alterations in genes related to liver steatosis [129]. As recently reviewed [123], male livers showed different TG levels and phospholipid compositions compared to their female counterparts, which may contribute to the higher susceptibility to NAFLD observed in males [130]. Following a similar experimental approach, Lomas-Soria and colleagues also reported that maternal obesity caused more severe hepatic changes in male F1 rats than in females [131]. Moreover, sex-specific differences have also been observed in adipose tissue following maternal obesity [132,133]. Dimorphism in hepatic lipid metabolism was studied using a transcriptomic approach in zebrafish, *Danio rerio*. Male livers were more likely to be sensitive to sex hormones than female livers [134]. Turola and colleagues [135] demonstrated that, after overfeeding, both young and old male zebrafish developed hepatic steatosis and fibrosis. The same outcome was observed in old female zebrafish, which also showed low E2 levels, but not in young female fish. Old and overfed female zebrafish exhibited a gene expression profile associated with hepatic fat accumulation similar to males. Specifically, all groups except the young female group revealed the upregulation of *pparγ* and *srebp1c*, markers involved in de novo lipogenesis, along with the downregulation of the cAMP-responsive element-binding protein (*creb313*), which plays a protective role in the development of steatosis [135]. E2 treatment performed a sex-specific regulatory role in hepatic cholesterol metabolism in zebrafish. In females, the upregulation of estrogen receptor 1 (*esr1*) after E2 treatment decreased the expression of hepatic 3-hydroxy-3-methylglutaryl-coenzyme A reductase *(hmgcr)*, a rate-limiting enzyme involved in cholesterol synthesis. In contrast, in males, the upregulation of *esr1* enhanced the expression of *hmgcr* and increased the production of hepatic cholesterol. Accordingly, low concentrations of E2 in male zebrafish promoted lipid deposition [136].

To better understand the pathophysiology of NAFLD, especially in terms of whether chronic inflammation induces metabolic changes in the liver and contributes to hepatic fat accumulation, an interleukin-6-overexpressing (I-OE) zebrafish model was used. The authors described the sex-specific control of the hepatic lipid profile. Male livers developed a higher concentration of saturated lipids and lower levels of unsaturated lipids. In contrast, increased levels of unsaturated lipids and a reduction in saturated lipids were found in the female I-OE livers, suggesting a potential defensive mechanism operating in females [137].

Interestingly, obesity, coupled with other metabolic diseases, is characterized by chronic inflammation and elevated levels of inflammatory chemokines and cytokines, such as monocyte chemoattractant protein-1 (MCP-1) and tumor necrosis factor-α (TNF-α), in adipose tissue, the liver, and/or blood. Additionally, there is increased accumulation and activation of macrophages and dendritic cells in the adipose tissue and liver [138]. Recent research has highlighted the role of the ECS in regulating food intake, energy metabolism, and inflammatory responses in the context of obesity [138,139,140,141]. However, most studies conducted on mice have predominantly focused on males, despite the emerging literature indicating that the energy balance is regulated differently in females and males. For example, in the case of male mice, it has been shown that CB1 activation can reduce food intake and inflammation, indicating that the ECS may be a potential target for combating obesity [142,143,144]. Mehrpouya-Bahrami and colleagues [145] demonstrated that treatment with SR141716A, a CB1 antagonist, in diet-induced obese C57BL/6 J male mice resulted in the mitigation of the obese phenotype. Furthermore, treatment with SR141716A attenuated inflammation, leading to lower levels of IL-17 and macrophage inflammatory protein-1α (MIP-1α) in adipose tissue [145].

### 4.3. Sex-Specific ECS Interactions with Metabolism

Over the years, it has been demonstrated that the ECS can regulate adipose β-oxidation, hepatic lipid and cholesterol biosynthesis in a sex-dependent manner, specifically in mammals. In addition, changes in nutritional programming due to maternal dietary restrictions during fetal development may play a significant role in the early-life programming of the ECS and the long-lasting effects of this on energy metabolism. This point will be now reviewed, and the main effects of maternal diets at the gene and protein levels are summarized in Figure 2.

Ramírez-López and co-workers [150] reported dissimilar transcriptomic profiles in male and female F1 rats from long-term calorie-restricted dams [150]. Interestingly, in female offspring, no significant alterations were observed in the hypothalamic region. In contrast, there was an upregulation of *Cnr1* and *Cnr2* expression in the hypothalamus of male F1 offspring fed with the restricted diet. The downregulation of *Nape-pld* was also detected in the liver of male F1 offspring, while females showed a decrease in the hepatic expression of *Cnr1*, *Faah*, and *Mgll*. Notably, even control animals exhibited sex differences in the hepatic gene expression of *Cnr1*, *Cnr2*, and *Mgll*.

When focusing on the perirenal adipose tissue, sex-specific differences were found in the gene expression of *Cnr1* and *Cnr2* in the calorie-restricted group. This also held for *Faah*, *Dagl*, and *Mgll* in the control group. Specifically, the calorie-restricted males exhibited downregulation in the genes coding for *Faah*, *Dagl*, and *Mgll*, while females showed the downregulation of *Nape-pld*. The downregulation of fatty acid β-oxidation markers, such as carnitine palmitoyltransferase 1b (*Cpt1b*) and acyl-CoA oxidase 1 (*Acox1*), along with the mitochondrial respiration marker cytochrome-c-oxidase subunit IV isoform 1 (*Cox4i1*) were reported in the calorie-restricted male perirenal adipose tissue, and a decrease in *Scd1* was noted in the calorie-restricted female perirenal adipose tissue. Additionally, the authors described dissimilar levels of *Scd1*, *Cpt1b*, and *Cox4i1* among males and females in the control group. Similar sex-based differences were also observed in the liver for the lipogenesis regulator *Acaca* (acetyl-CoA carboxylase alpha) and the cholesterol metabolism marker insulin-induced gene 1 (*Insig1*). The downregulation of genes related to lipogenesis (*Acaca*, fatty acid synthase [*Fasn*]), cholesterol metabolism (*Insig1*, *Hmgcr*), and fatty acid β-oxidation (carnitine palmitoyltransferase 1a [*Cpt1a*], *Acox1*) was evident in the calorie-restricted female liver. This study does not only describe sexual dimorphism in the ECS but also in ECS-related mechanisms such as lipid metabolism and feeding behavior [143].

Gestational exposure to high-calorie diets in rat offspring also highlighted long-term modifications in the ECS, with sex-specific differences in the offspring. Male offspring from dams on a free-choice diet consisting of a mixture of chocolates and standard chow exhibited lower intake of standard food after an acute dose of AM251, a CB1 receptor inverse agonist. In contrast, male pups from the control group (fed on standard chow only) showed lower chocolate consumption without changes in their intake of standard chow. The study further analyzed gene expression profiles in the brain, liver, and adipose tissue, focusing on the metabolic pathways of the ECS [150]. Similarly, other studies reported sex-specific changes in the ECS of offspring following HFD exposure. Both male and female rats evidenced increments in fat mass and body weight, though only male adult offspring displayed hyperleptinemia and higher energy expenditure, effects which were potentially linked to the downregulation of hypothalamic leptin signaling. Interestingly, hypothalamic CB1 protein level was upregulated in male pups, whereas female offspring exposed to maternal HFD showed elevated CB2 levels at birth in the same region [147]. A recent study by Miralpeix’s group further explored the effects of diet-induced obesity on the hypothalamic ECS over two temporal phases. Short-term (7-day) HFD exposure led to elevated levels of 2-AG in both male and female mice, accompanied by an overexpression of enzymes involved in endocannabinoid synthesis. In contrast, long-term HFD (90-day) treatment returned the levels of the endocannabinoids to the baseline. In female mice, elevated hypothalamic endocannabinoids were linked to the activation of thermogenesis, delaying the onset of obesity and contributing to milder symptoms. This reinforces the potential role of the ECS in regulating obesity progression, although the exact mechanisms remain unclear [151].

When investigating the effects of perinatal HFD exposure, sex-specific long-term changes in hepatic ECS components were observed in adult F1 rats [101]. Maternal HFD induced the upregulation of the hepatic proteins CB1, CB2, FAAH, and MGLL in male offspring, with the upregulation of the latter marker also noted in female offspring. Additionally, increased hepatic ERα and elevated TG levels were observed in male offspring. In contrast, the expression of ERα and TG levels were not modified in female livers. Miranda and colleagues correlated the levels of ECS components and ERα with the presence of the estrogen response element described in the *Faah* sequence [34]. Using a similar approach as Miranda and co-workers (2018) [101], Fassarella reported that maternal HFD also increased hepatic CB1 in males while decreasing the CB2 and AEA tone in female rats. Based on their previous studies, the authors proposed that AR had a stimulatory effect on CB1 expression in the neonate HFD male offspring. Furthermore, markers of lipid metabolism were also altered in a sex-specific manner in the liver. *Apob* (Apolipoprotein B), which is relevant for the hepatic TG exportation via very-low-density lipoprotein (VLDL), as well as *Pparα* and *Cpt1b*, were downregulated in male F1 offspring, while *Srebf1c* and *Acaca* were upregulated in female F1 offspring. Overall, these results suggest that, despite the fact that sex-specific regulation and the participation of the ECS in lipid metabolism are still poorly characterized, maternal nutrition can modulate the ECS of the offspring, predisposing subjects to or preventing the development of metabolic diseases [148]. Although both studies focus on HFD-exposed animals, the different timing and duration of HFD treatment may lead to discrepant results. In Miranda’s study, HFD exposure occurred from the pre-mating period to weaning, whereas Fassarella’s study administered the HFD during pre-mating and pregnancy [101,148]. Moreover, the timing of F1 measurements varied between studies, which potentially influenced ECS modulation and metabolic outcomes. These timing variations could be crucial in determining the presence of specific metabolic adaptations or long-term effects in offspring. We found a similar situation when examining the studies from [146,152], which showed different CB1 profiles in WAT at postnatal day 21 and during adulthood. In addition, De Almeida demonstrated that maternal HFD modulated ECS signaling and lipid metabolism in a sex-specific manner in WAT at weaning and in adult rat offspring. At postnatal day 21, CB1, CB2, FAAH and MGLL showed different expression patterns in male and female pups along with adipogenic and lipogenic markers (PPARγ, ACC, CCAAT/enhancer binding protein α [CEBPα]) in WAT. During adulthood, HFD increased *Cnr1* and CB1 levels in the visceral and subcutaneous WAT of female rats. In OVX animals, *Cnr1* expression was downregulated, while CB1 levels were increased. This disparity suggests potential post-transcriptional or post-translational mechanisms regulating ECS components in response to maternal HFD exposure. Additionally, maternal HFD caused a reduction in plasma E2 levels in female F1 offspring associated with an increase in ERα binding to the *Cnr1* promoter [146,152]. On the other hand, female offspring exposed to maternal undernutrition exhibited lower levels of *Cnr1* in subcutaneous WAT, while in males, *Faah* expression was deregulated. This study also provided evidence of sex-different transcriptomic profiles for *Faah*, *Daglα*, *Daglβ*, and *Mgll* [150].

Finally, reduced CB1 and increased FAAH were observed in the BAT of male newborn pups from HFD-exposed dams. This suggests the activation of BAT associated with energy expenditure and the existence of a protective mechanism against the concentration of endocannabinoids, respectively. In contrast, higher expression of CB2 and MGLL were reported in the BAT of female offspring from HFD dams. The role of CB2 in the peripheral control of energy metabolism is still poorly understood, but this upregulation in female pups may indicate a greater macrophage infiltration in the BAT. The CB2 receptor is mainly found in immune cells and the lack of CB2 leads to a reduction in infiltrating macrophages and lower levels of pro-inflammatory chemokines in the mouse brain. Therefore, indicating a potential involvement of CB2 in the modulation of inflammation and mitigating the negative metabolic consequences of maternal HFD exposure, which predisposes individuals to adult hyperphagia and obesity [147].

## 5. ECS Vulnerability Factors and ECS-Related Fingerprints

As mentioned above, many studies have demonstrated notable sex differences in endocannabinoid signaling and its impact on metabolism. These differences highlight the need to consider sex-specific responses to the exposome, mediated through endocannabinoid modulation.

### 5.1. The Sex-Specific Effects of EDCs on the ECS Liver Metabolism

Endocrine-disrupting chemicals (EDCs) are compounds that interfere with the synthesis, transport, binding, and/or elimination of endogenous hormones [153]. It is noteworthy that, as stated in Section 2, estrogen can modify the expression of *Faah* since its promoter harbors an estrogen-responsive element [35], and sex steroids regulate *Cnr1* expression [38]. Consequently, exposure to substances that mimic or antagonize steroid hormones has been shown to disrupt the ECS of different species, thus affecting neuronal functions, reproductive performance, and hepatic metabolism in a sex-dependent manner. These effects will be further examined across different animal species.

In males, mice perinatal exposure to 10 µg/mL bisphenol A (BPA) decreased the expression of *Cnr1* in the hypothalamus, a change likely associated with the activation of anorexigenic signals and, therefore, altered food intake [154]. Likewise, the negative impact of both 4000 µg/kg body weight (bw)/day BPA and 1500 µg/kg bw/day Di-isononyl phthalate (DiNP) on appetite was also reported in gilthead sea bream (*Sparus aurata*) males, along with the downregulation of genes coding for *cnr1* and *npy* as well as the downregulation of AEA levels [155]. By using mutant animals (CB1 KO mice) and drugs to enhance the ECS, it has been demonstrated that the neurotoxicity triggered by exposure to insecticides in male mice, especially to organophosphorus (OP), is mediated by the alterations caused in the endocannabinoid receptors and/or the enzymes involved in endocannabinoid metabolism [156,157]. Moreover, in male mice, the OP-induced MGLL inhibition increased the levels of 2-AG, associated with impaired animal motility. Likewise, the inhibition of FAAH in males exposed to different OPs led to the accumulation of AEA and mice hypomotility [158]. The activity of these two enzymes was inhibited in the liver of male mice exposed to 2 mg/kg of the OP chlorpyrifos (CP), but this was only the case in HFD animals, suggesting that obesity could be a risk factor for increased CPs hepatotoxicity [159]. In zebrafish, male exposure to both 500 μg/L BPA and tetrabromobisphenol (TBBPA) significantly altered the hepatic metabolism, induced liver steatosis and obesity, and increased appetite signals by activating the transcription of *cnr1* [160].

In females, liver steatosis resulting from zebrafish exposure to 100 µg/L BPA was linked to increased hepatic levels of 2-AG and AEA, but decreased levels of POEA. The hepatic overexpression of *cnr1* and the dysregulation of the enzymes in charge of the endocannabinoid metabolism were also reported [161]. Similar results were observed in zebrafish females exposed to lower doses of BPA, especially in the micrograms per liter range (µg/L), in which the increased hepatic deposition of lipids was concomitant with changes in the expression of ECS enzymes and, consequently, with changes in the levels of endocannabinoids. However, in this case, neither the appetite molecular markers nor the food intake was affected by BPA exposure [162]. In contrast, the liver steatosis reported when exposing zebrafish females to 4.2 µg/L DiNP was not only related to changes in the expression of *cnr1*, *dagl*, *mgll*, and lipid metabolism, but also to an increase in the orexigenic signals [163]. In mice, the F1 dams exposed to 0.1 mg/kg of the flame-retardant DE-71, a polybrominated diphenyl ether (PBDE), displayed altered liver levels of AEA and related fatty acid-ethanolamides, docosahexaenoyl ethanolamide (DHEA), and *N*-oleoylethanolamide (OEA), even though no changes were reported in F0 females [164]. Altogether, these findings underscore the importance of sex-specific research into the intricate interplay between environmental contaminants and endocannabinoid-dependent metabolic pathways.

### 5.2. Sex Response to Exogenous Cannabinoids

Exogenous cannabinoids or exocannabinoids encompass both natural and synthetic cannabinoids. Natural compounds derived from the cannabis plant are referred to as phytocannabinoids, distinguishing them from synthetic cannabinoids and endocannabinoids. The constituents found in *Cannabis sativa* are characterized by remarkable diversity, comprising a total of 565 natural compounds. Among these, 120 distinct cannabinoids have been identified and categorized into 11 general types. Notably, the most renowned and extensively researched cannabinoids are THC, recognized as the primary psychoactive element, and cannabidiol (CBD), acknowledged as a predominant non-psychotropic constituent of *Cannabis sativa* [165].

The role of exocannabinoids in liver health is complex and multifaceted. The chronic use of exocannabinoids, particularly THC, has been linked to adverse effects in terms of liver function, including potential implications for metabolic disorders and liver diseases. Understanding how these effects manifest differently in males and females is crucial for designing tailored interventions. It is noteworthy that sex differences in how individuals respond to cannabinoids seem to stem from the influence of sexual hormones, with E2 being pinpointed as the primary hormone responsible for the different effects of cannabinoids seen in adults [166]. One factor contributing to the possibility of THC having more pronounced effects in females compared to males is its metabolism in the liver. The metabolic process of THC differs between male and female rats. In females, there is a preference for metabolizing THC into 11-hydroxy-Δ^9^-tetrahydrocannabinol (11-hydroxy-THC), which is a very potent analgesic; in males, THC is metabolized into 11-hydroxy-THC as well as various other compounds, most of which are inactive [167]. This difference is primarily due to female rodents exhibiting higher levels of hepatic cytochrome P-450 isozymes and aldehyde oxygenase activity, which transform THC into 11-hydroxy-THC [168].

Despite all the above-mentioned differences, the studies focused on the impact of exocannabinoids on metabolism have been mainly conducted in males, neglecting the impact of these compounds on female metabolic activity. The prolonged oral consumption of THC (10 mg/kg), but not CBD (30 mg/kg), in male mice alleviated diet-induced obesity by improving glucose tolerance and liver steatosis, and by decreasing adipocyte hypertrophy [169]. Likewise, the same authors proved that THC-enriched medical oil prevented weight gain and ameliorated diet-induced liver steatosis, whereas CBD-enriched medical oil increased this last parameter in male mice [170]. On the contrary, neither the oral treatment of male mice with THC nor CBD (at 2.5 and 2.39 mg/kg, respectively) was able to reduce liver steatosis derived from HFDs, but CBD treatment improved glucose tolerance and reduced liver inflammation [171]. Similarly, treatment with 5 mg/kg CBD was reported to exert anti-inflammatory effects in the livers of male mice with non-alcoholic steatosis induced by a high-fat high cholesterol diet [172].

## 6. Conclusions

To summarize, this review highlights the crosstalk between the ECS and sex steroids, with the endocannabinoids modulating sex steroid levels and gonadotropins, and the ECS itself being strongly influenced by sexual hormones. This dynamic interaction has significant implications for metabolic health, particularly concerning sex-specific vulnerabilities to conditions such as obesity, liver diseases such as the NAFLD, and diabetes. Sex hormones appear to bidirectionally regulate the ECS, leading to tissue-specific and time-dependent metabolic effects. However, the variability of results across animal studies, combined with the relative scarcity of research on female models, limits our ability to fully grasp the sex-specific roles of the ECS. As such, future research should focus on standardizing experimental approaches to better elucidate how hormonal fluctuations across the lifespan—such as during puberty, pregnancy, and menopause—affect ECS-mediated metabolic outcomes. Moreover, the use of gonadectomized animals, as already implemented in several studies, is a well-known strategy to avoid interference from endogenous sex steroid production.

Future studies should also aim to elucidate the exact mechanisms by which maternal nutrition influences ECS-mediated appetite control and fat storage in offspring, and how these effects may differ between males and females. This could unlock new strategies for addressing obesity and metabolic disorders, which may be tailored to the specific needs of different sexes. Additionally, the growing prevalence of EDCs and the use of exogenous cannabinoids underscores the need to explore how these chemicals impact the ECS differently in males and females. Research in this area could inform the development of more personalized therapeutic strategies that take sex into account as a critical biological variable in metabolic health.

## Figures and Tables

**Figure 1 ijms-25-11909-f001:**
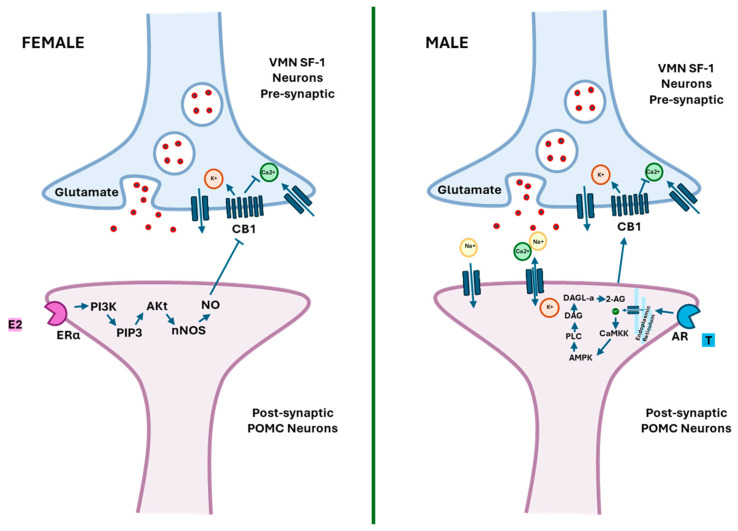
A schematic illustration summarizing the endocannabinoid signaling occurring at VMN SF-1/ARC POMC synapses in females and males. In females, estrogen binding to ERs increases the activity of PI3K and the production of PIP3, which upregulates Akt and stimulates nNOS. Endocannabinoid levels in the synaptic cleft are then reduced by nNOS through the inhibition of their synthesis, the improvement of their catabolism, and the promotion of their reuptake. Moreover, the NO produced may act in a retrograde fashion to uncouple CB1 from its ligands in the glutamatergic nerve terminal. On the other hand, testosterone in males activates a putative membrane AR and raises the levels of intracellular calcium. A calcium increase activates CaMKK and the AMPK complex, improving the production of 2-AG by DAGL-α. The 2-AG release can retrogradely activate presynaptic CB1 in the VNM SF-1 neurons and inhibit calcium entry, ultimately decreasing glutamatergic input. VMN: ventromedial nucleus; SF-1: steroidogenic factor-1; ARC: arcuate nucleus; POMC: proopiomelanocortin; PI3K: phosphatidylinositol-3-kinase; PI3P: phosphatidylinositol-3-phosphate; nNOS: nitric oxide synthase; NO: nitric oxide; CaMKK: calmodulin-dependent protein kinase; AMPK: AMP-activated protein kinase; Akt: protein kinase B; ESRɑ: estrogen receptor ɑ; E2: estradiol; AR: androgen receptor; 2-AG: 2-arachidonoylglycerol; PLC: phospholipase C; DAGL: diacylglycerol lipases; CB1: cannabinoid receptor.

**Figure 2 ijms-25-11909-f002:**
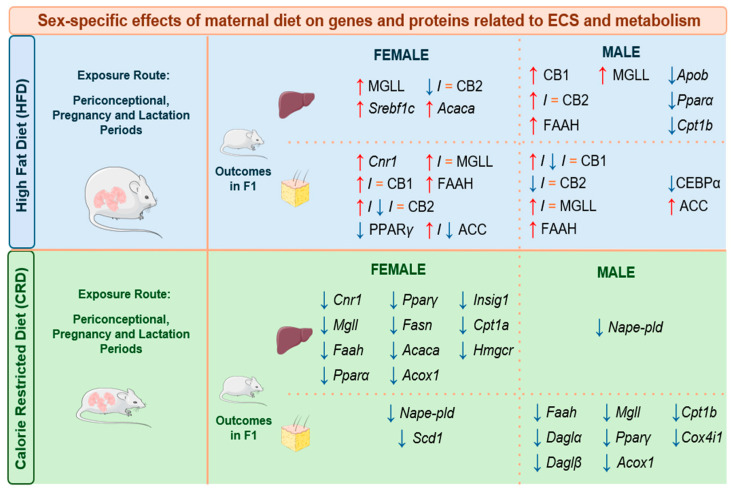
A summary of the sex-specific effects of a high-fat diet and calorie-restricted diet on gene expression and protein levels related to the ECS and metabolism in F1 mice. Dams were treated before pregnancy, during pregnancy and/or lactation. Cases when a treatment induced a significantly increased response are marked with a red “↑”; a significantly decreased response is marked with a blue “↓”; and no change is marked with a “=”. Cases with contradictory results found between studies are marked as “↑/=”, “↓/=”, or “↑/↓/=”. Readouts extracted from the bibliography are written in capital letters for protein levels, and in italics for gene expression [101,146,147,148,149]. Abbreviations: CB1/*Cnr1:* cannabinoid receptor 1; CB2: cannabinoid receptor 2; MGLL/*Mgll:* monoacylglycerol lipase; FAAH/*Faah:* fatty acid amide hydrolase 1; *Nape-pld: N*-acyl-phosphatidylethanolamine-hydrolysing phospholipase D; *Srebf1c:* sterol regulatory element binding factor-1c; *Acaca:* acetyl-coA carboxylase alpha; *Acox1:* acyl-CoA oxidase 1; *Apob:* apolipoprotein B; *Hmgcr:* 3-hydroxy-3-methylglutaryl-coenzyme A reductase; *Ppar:* peroxisome proliferator-activated receptor; *Cpt1:* carnitine palmitoyltransferase 1; *Scd1*: stearoyl-CoA desaturase; *Cox4i1:* cytochrome c oxidase subunit 4 isoform 1; ACC: acetyl coenzyme A carboxylase; CEBPα: CCAAT/enhancer binding protein α.

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
