# Peer review of "Endocannabinoid System and Metabolism: The Influences of Sex"

_ijms, 2024, doi:10.3390/ijms252211909_

Round 1
Reviewer 1 Report
Comments and Suggestions for Authors
The authors aimed to review literature on sex differences in the effects of the endocannabinoid system (ECS) on metabolic processes. The review is informative and comprehensive, providing a good overview of the current state of the field. However, I have several suggestions for improvement.
A similar topic was previously reviewed by Wagner (Wagner E. J. (2016). Sex differences in cannabinoid-regulated biology: A focus on energy homeostasis. Frontiers in Neuroendocrinology, 40, 101–109. https://doi.org/10.1016/j.yfrne.2016.01.003), and there is significant overlap in content, particularly regarding brain effects and the female section of Figure 1, which should be revised. I also suggest citing this paper in the reference list.
Additionally, I recommend adding more synthesis and future perspectives to the discussion. While some studies are described in detail, a clearer summation of findings and their implications for future research would enhance the manuscript. Furthermore, the review would benefit from additional figures, particularly on the effects of the ECS in the liver and adipose tissue, to provide a more comprehensive visual representation of the discussed topics.
Regarding the references, I noticed that they are not prepared according to the journal’s guidelines. Some references are missing (e.g., Struik et al., 2013), and there are inconsistencies in the abbreviations of journal names. Additionally, some citations are incomplete (e.g., 88, 96), though the authors may already be aware of these issues.
There are also a few minor points to address:
- "endocannabinoid transporters to cross the cell membrane" – Endocannabinoids are lipids that readily cross the lipid membrane. The role of transporters in this context should be clarified.
- "stimulation of mitogen-activated protein kinase" – There are three main types of MAPKs. Either make this plural or specify which one is involved.
- "This increase in food intake hampers the signalling pathway involving Phosphoinositide 3-kinases (PI3K), which produce phosphatidylinositol triphosphate (PIP3) and Protein kinase B (Akt) in POMC neurons" – PI3K activates Akt, rather than producing it.
- "proopiomelanocortin, the precursor of α-MSH and β-endorphin in POMC neurons" – "Proopiomelanocortin" is the same as POMC, so this could be reworded for clarity.
- "Even though the downregulation of deiodinase 2 (Dio2) and the increase of local Triiodothyronine (T3) levels, which has the potential to stimulate uncoupling protein 1 (UCP-1), also known as thermogenin, this signal remained unaltered, resulting in the absence of thermogenesis deregulation" – This sentence is unclear and needs revision for better understanding.
- "In parallel, FAAH and MAGL dropped in males and females, respectively" – It would be helpful to specify if the observed effects are mediated via CB1 receptors, as this would clarify any apparent contradictions with the CB2-related text on page 6.
The manuscript would benefit from some English editing to improve clarity in certain sections. However, overall, it is written in a clear and understandable manner.
Author Response
Reviewer 1
Open Review
The authors aimed to review literature on sex differences in the effects of the endocannabinoid system (ECS) on metabolic processes. The review is informative and comprehensive, providing a good overview of the current state of the field. However, I have several suggestions for improvement.
A similar topic was previously reviewed by Wagner (Wagner E. J. (2016). Sex differences in cannabinoid-regulated biology: A focus on energy homeostasis. Frontiers in Neuroendocrinology, 40, 101–109. https://doi.org/10.1016/j.yfrne.2016.01.003), and there is significant overlap in content, particularly regarding brain effects and the female section of Figure 1, which should be revised. I also suggest citing this paper in the reference list.
We apologise for this oversight. The review from Wagner 2016 was cited in the manuscript and the results integrated with recent studies regarding the ECS regulation of the appetite at the brain level in males (Conde et al., 2017; Fabelo et al., 2018). Wagner 2016 was also cited in the caption of Figure 1.
Additionally, I recommend adding more synthesis and future perspectives to the discussion. While some studies are described in detail, a clearer summation of findings and their implications for future research would enhance the manuscript. Furthermore, the review would benefit from additional figures, particularly on the effects of the ECS in the liver and adipose tissue, to provide a more comprehensive visual representation of the discussed topics.
We have amended the conclusions including final considerations for all the sections and emphasizing how future research could address the challenges associated with the topics discussed. Additionally, we would like to highlight that the role of the ECS in liver and fat tissue has already been summarized in Figure 2, which details the effects of treatments administered to dams before pregnancy, during pregnancy, and throughout lactation on liver and fat tissue on the ECS.
Regarding the references, I noticed that they are not prepared according to the journal’s guidelines. Some references are missing (e.g., Struik et al., 2013), and there are inconsistencies in the abbreviations of journal names. Additionally, some citations are incomplete (e.g., 88, 96), though the authors may already be aware of these issues.
The reference for Struik et al 2018 has been added. The bibliography has been revised and all the inconsistencies (i.e. abbreviations of journal names) have been corrected.
There are also a few minor points to address:
- "endocannabinoid transporters to cross the cell membrane"– Endocannabinoids are lipids that readily cross the lipid membrane. The role of transporters in this context should be clarified.
This part has been modified.
- "stimulation of mitogen-activated protein kinase"– There are three main types of MAPKs. Either make this plural or specify which one is involved.
We have made it plural as the endocannabinoids can activate different MAPKs.
- "This increase in food intake hampers the signalling pathway involving Phosphoinositide 3-kinases (PI3K), which produce phosphatidylinositol triphosphate (PIP3) and Protein kinase B (Akt) in POMC neurons"– PI3K activates Akt, rather than producing it.
The sentence has been revised.
- "proopiomelanocortin, the precursor of α-MSH and β-endorphin in POMC neurons"– "Proopiomelanocortin" is the same as POMC, so this could be reworded for clarity.
We overlooked this abbreviation, “proopiomelanocortin” has been now reworded as POMC.
- "Even though the downregulation of deiodinase 2 (Dio2) and the increase of local Triiodothyronine (T3) levels, which has the potential to stimulate uncoupling protein 1 (UCP-1), also known as thermogenin, this signal remained unaltered, resulting in the absence of thermogenesis deregulation"– This sentence is unclear and needs revision for better understanding.
The sentence has been revised.
- "In parallel, FAAH and MAGL dropped in males and females, respectively"– It would be helpful to specify if the observed effects are mediated via CB1 receptors, as this would clarify any apparent contradictions with the CB2-related text on page 6.
The sentence was revised.
Dias-Rocha et al., 2023 reported that the reduced levels of FAAH were associated with an increase in both CB1 and CB2 in the nucleus accumbens of male rats from HFD mothers, which could be related to the decreased levels of AEA and 2-AG in HFD mother milk. Females from HFD mothers only displayed decreased MAGL levels without alterations in CB1 and CB2. Thus, in males, the alteration of FAAH could be potentially mediated by CB1/CB2.
On page 6, the role of CB1 and CB2 on appetite was reported. Since in Dias-Rocha et al., 2023, male offspring from HFD mothers showed an increased preference for HFD along with the CB1 increase in the nucleus accumbens, despite still showing an increase also of CB2, the effect of CB1 on appetite possibly prevailed leading to this food preference considering that the effects of these two receptors on appetite are the opposite, with CB1 stimulating it and CB2 decreasing the appetite instead.
Comments on the Quality of English Language
The manuscript would benefit from some English editing to improve clarity in certain sections. However, overall, it is written in a clear and understandable manner.
English has been revised and some parts are now rewritten.
Submission Date
07 October 2024
Date of this review
16 Oct 2024 15:41:28
Reviewer 2 Report
Comments and Suggestions for Authors
The authors wrote very interesting review about sexual dimorphism of the ECS in different animal models, providing evidence about the crosstalk between endocannabinoids and sex hormones in different metabolic pathways.
The abstract is clearly written, the introduction is well and comprehensively written,
references are updated, subheadings with gender differences are well described. The conclusion is clearly written.
The figures are nice, it's just necessary to explain all the abbreviations in the legends.
Author Response
Reviewer 2
Open Review
Comments and Suggestions for Authors
The authors wrote very interesting review about sexual dimorphism of the ECS in different animal models, providing evidence about the crosstalk between endocannabinoids and sex hormones in different metabolic pathways.
The abstract is clearly written, the introduction is well and comprehensively written, references are updated, subheadings with gender differences are well described. The conclusion is clearly written.
The figures are nice, it's just necessary to explain all the abbreviations in the legends.
We thank the reviewer for this thoughtful and positive feedback on our manuscript. We have considered this suggestion and added all abbreviations in the figure legends.
Submission Date
07 October 2024
Date of this review
15 Oct 2024 20:48:37
Reviewer 3 Report
Comments and Suggestions for Authors
The authors of this manuscript aim to review the sexual dimorphism of the endocannabinoid system (ECS) in different animal models, providing evidence about the crosstalk between endocannabinoids and sex hormones in various metabolic pathways. Additionally, their goal is to underscore the importance of understanding how endocrine-disrupting chemicals and exogenous cannabinoids influence ECS-dependent metabolic pathways in a sex-dependent manner.
The reviewer congratulates the authors on this excellent summary of the most recent knowledge in this area. The presented evidence is comprehensive and will serve as a reference paper in the future. The reviewer suggests adding a figure showing the general sex differences in the ECS. The authors might also consider adding a paragraph on the potential consequences of sexual differences in the ECS for the outcome of severe inflammatory/infectious conditions such as sepsis, keeping in mind that the impact of obesity on sepsis outcome is controversial in the literature (obesity paradox).
Comments on the Quality of English LanguageThe manuscript's flow is easy to follow, and the reviewer has no concerns about the quality of the English language.
Author Response
Reviewer 3
Open Review
Comments and Suggestions for Authors
The authors of this manuscript aim to review the sexual dimorphism of the endocannabinoid system (ECS) in different animal models, providing evidence about the crosstalk between endocannabinoids and sex hormones in various metabolic pathways. Additionally, their goal is to underscore the importance of understanding how endocrine-disrupting chemicals and exogenous cannabinoids influence ECS-dependent metabolic pathways in a sex-dependent manner.
The reviewer congratulates the authors on this excellent summary of the most recent knowledge in this area. The presented evidence is comprehensive and will serve as a reference paper in the future. The reviewer suggests adding a figure showing the general sex differences in the ECS.
The authors would like to thank the reviewer for the encouraging feedback. We also thank you for your suggestion to include a figure summarising the general sex differences in the ECS. Due to the complexity and variability across different studies, methodologies, and animal models, creating such a figure poses significant challenges. The controversy surrounding some findings, particularly concerning sex-specific ECS effects, would make it difficult to produce a clear, unified representation.
The authors might also consider adding a paragraph on the potential consequences of sexual differences in the ECS for the outcome of severe inflammatory/infectious conditions such as sepsis, keeping in mind that the impact of obesity on sepsis outcome is controversial in the literature (obesity paradox).
Thank you for your thoughtful suggestion regarding the inclusion of a paragraph on the implications of sexual differences in the ECS for severe inflammatory or infectious conditions such as sepsis. While we recognize the importance of this topic, our review primarily focuses on metabolic regulation and sex-specific differences in the ECS, and we think that a detailed discussion of sepsis falls outside the scope of this manuscript. However, we greatly appreciate your insight and have incorporated additional information on sex differences in the ECS's role in controlling inflammatory responses in the context of obesity at the end of section 4.2. We hope this addition enhances the depth of the review and enriches the discussion of the ECS in these related conditions.
Comments on the Quality of English Language
The manuscript's flow is easy to follow, and the reviewer has no concerns about the quality of the English language.
Submission Date
07 October 2024
Date of this review
16 Oct 2024 22:18:10
Round 2
Reviewer 1 Report
Comments and Suggestions for Authors
The authors have implemented revisions that improve the manuscript, making it suitable for publication.
Minor Issues:
-
The section on endocannabinoid transporters still requires further improvement for clarity and completeness
-
Please revise the phrase “This point will now be reviewed” in the text preceding Table 2
Author Response
Comments and Suggestions for Authors
The authors have implemented revisions that improve the manuscript, making it suitable for publication.
Minor Issues:
The section on endocannabinoid transporters still requires further improvement for clarity and completeness
The authors thank the reviewer for the valuable comment, which has contributed to improving the manuscript’s quality. We have revised the relevant section accordingly and highlighted the changes in yellow. However, we did not investigate deeply into this topic, as it is not the primary focus of the review, and we aim to keep the introduction concise. Additionally, recent comprehensive reviews on endocannabinoid transport (10.1007/978-1-0716-2728-0_1; 10.1016/bs.vh.2014.12.011) have been cited in the manuscript to provide completeness.
Please revise the phrase “This point will now be reviewed” in the text preceding Table 2
The sentence has been revised according to the reviewer comment.
Submission Date
07 October 2024
Date of this review
28 Oct 2024 12:18:04